# A Model of Lipid Monolayer–Bilayer Fusion of Lipid Droplets and Peroxisomes

**DOI:** 10.3390/membranes12100992

**Published:** 2022-10-13

**Authors:** Maksim A. Kalutsky, Timur R. Galimzyanov, Rodion J. Molotkovsky

**Affiliations:** 1A.N. Frumkin Institute of Physical Chemistry and Electrochemistry, Russian Academy of Sciences, 31/5 Leninskiy Prospekt, 119071 Moscow, Russia; 2Department of Theoretical Physics and Quantum Technologies, National University of Science and Technology “MISiS”, 4 Leninskiy Prospekt, 119049 Moscow, Russia

**Keywords:** lipid droplet, peroxisome, membrane fusion, membrane elasticity, molecular dynamics

## Abstract

Lipid droplets are unique organelles that store neutral lipids encapsulated by the lipid monolayer. In some processes of cellular metabolism, lipid droplets interact with peroxisomes resulting in the fusion of their envelopes and the formation of protrusions of the peroxisome monolayer, called pexopodia. The formation of pexopodia is facilitated by free fatty acids generated during lipolysis within lipid droplets. In this work, we studied the fusion of monolayer and bilayer membranes during the interaction between lipid droplets and peroxisomes. To this end, we built the energy trajectory of this process using the continuum elasticity theory and investigated the molecular details of the fusion structures utilizing molecular dynamics. We divided the fusion process into two stages: formation of a stalk and its consequent expansion into pexopodia. We found that in the considered system, the stalk was energetically more stable and had a lower energy barrier of formation compared to the case of bilayer fusion. The further evolution of the stalk depended on the value of the spontaneous curvature of the membrane in a threshold manner. We attributed the possible expansion of the stalk to the incorporation of free fatty acids into the stalk region. The developed model allowed describing quantitatively the process of monolayer–bilayer fusion.

## 1. Introduction

Eukaryotic cells comprise many interacting organelles. This interaction often leads to the contact and restructuring of lipid membranes, which enclose organelles and perform a barrier function. One form of membrane reshaping processes is their fusion, i.e., combining the material of the membranes themselves and the water volumes surrounded by them. Membrane fusion plays a critical role in a plethora of biological processes such as exocytosis, fertilization, secretion, synaptic transmission, etc. [1].

According to the commonly accepted paradigm, bilayer membrane fusion occurs in several stages via local non-stationary contacts between interacting leaflets and leads to their structural reorganization [2,3,4]. At the initial stage, local contact between the membranes results in the fusion of proximal monolayers and the formation of an hourglass-shaped structure called a stalk [5,6,7]. Radial expansion of the stalk leads to the tail-to-tail contact of the distal leaflets and the formation of the so-called hemifusion diaphragm [4,8,9], consisting of a bilayer separating bulk volumes of organelles. The formation of a pore in the hemifusion diaphragm completes the fusion process [4,10,11]. The transient nature of fusion intermediates hinders their direct experimental observation, while theoretical considerations provide an effective way of their studying. For fusion to occur, the membranes need to deform and locally overcome the hydration repulsion between them [2]. The theoretical approach makes it possible to determine the energy required for this restructuring and allows calculating the energy barrier needed to be overcome for the formation of metastable fusion structures [12,13]. According to the previous estimates, the energy barrier for stalk formation equals several tens of *k_B_T*, depending on the model and starting conditions of fusion [14]. Energy barriers for the formation of the hemifusion diaphragm and fusion pore are of the same order of magnitude [13]. These barriers are overcome partly by the action of specific proteins known as “fusion proteins”, which bring fusing membranes at a small distance and keep them together [15,16]. The height of the energy barrier for stalk formation determines the waiting time for monolayer fusion which makes it the crucial parameter of this process [12,17]. Composition of the membrane also affects this energy barrier and, consequently, the stability of the stalk intermediate [18]; the mutual influence of fusion proteins and membrane composition determines the overall energy landscape of the fusion process [19].

Most cellular organelles have bilayer shells. In cells, there are unique assemblies, covered by the lipid monolayer—the so-called lipid droplets, which store neutral lipids, such as triolein and sterol esters [20]. The monolayer shell of phospholipids acts as a surfactant and prevents the coalescence of the oily core of lipid droplets. Recent studies suggest that lipid droplets are dynamic organelles and interact with almost all cellular compartments [21,22]. This interaction, as a rule, is reduced to the formation of a membrane contact site, which is facilitated by special proteins called tethering proteins [23,24]. These proteins keep membranes of interacting organelles at a small distance (~10 nm), allowing the protein-mediated exchange of lipids and other small molecules between them [24]. Although the formation of membrane contact sites does not typically lead to membrane fusion, existing evidence suggests that the interaction of lipid droplets with each other, as well as with the endoplasmic reticulum and peroxisomes can include the physical connection of the membranes [22]. In these cases, membranes of interacting organelles can form hemifusion structures such as lipidic bridges in the case of the ER, or even fuse completely with other organelles [24,25].

The contact of a lipid droplet with a peroxisome, an organelle responsible for the oxidation of free fatty acids (FFAs), is the least understood. Available experimental data [26] indicate that during fusion with a lipid droplet, peroxisomes can form specific structures, called pexopodia, representing lipid monolayers, invaginated into the lipid droplet’s core (see Figure 1). These structures have unique features and arise under specific conditions. According to Binns and coworkers [26], pexopodia were observed predominantly under the deficiency of a hydrocarbon source in cells grown on oleate nutrition. The emergence of such structures can be associated with the formation of free fatty acids in the core of the lipid droplet. This suggestion is indirectly confirmed by the accumulation of enzymes involved in fatty acids beta-oxidation in the pexopodia proximity in peroxisomes [26]. Localization of beta-oxidizing enzymes near the monolayer membrane separating the peroxisome from the lipid droplet hints that FFAs need to diffuse through this monolayer for oxidation to occur. If peroxisomes are deficient in oxidizing enzymes, they can form tubular structures named “gnarls”. These structures protrude deep into the lipid droplet core and are thought to be composed of FFAs accumulated at the pexopodia surface and unable to be oxidized.

Several key questions remain unanswered related to the fusion of lipid droplets with peroxisome membranes. Firstly, it is not clear what role hypothetical tethering proteins play in this fusion. Depending on the height of the energy barrier for stalk formation, they may be more or less involved in the fusion process; if the barrier height is small and the stalk is energetically stable intermediate, fusion may occur even without significant protein assistance. For example, the proteins may simply position membranes at a fixed distance between each other without the need for any further action. Secondly, the role of FFAs in the formation of pexopodia is still puzzling. Experimental data allow suggesting FFAs modify the monolayer separating pexopodia from lipid droplets, easing the formation of these structures and making them more stable [26]. However, the exact mechanism of this modification is still unknown. Monolayer–bilayer fusion itself has not been studied yet and comprises a challenging problem: when investigating this process, one needs to describe the geometrically and physically non-trivial contact of two monolayers and one bilayer (see Figure 1b). Current work aims to consider these issues and to build a model describing monolayer–bilayer fusion. By analogy with the bilayer fusion model, we expressed features of different stages of monolayer–bilayer fusion through several distinct physical parameters. These parameters characterized physico-chemical conditions that regulate the metabolism of lipid droplets in the cell. We generalized the classical bilayer fusion theory to the monolayer–bilayer process and built a complete trajectory of the process, starting with the contact of membranes and ending with the formation of the pexopodium. This made possible the determination of energy barriers for the formation of possible intermediate structures of this process. Calculation of energy barriers allowed us to address the stability of intermediate structures and the likelihood of their formation. In this work, we considered two such structures: the monolayer–bilayer stalk and the so-called π-shaped structure (see Figure 1b), which results from the stalk expansion and sagging of the inner peroxisome monolayer into the lumen between the fused contact monolayers [27].

For calculations, we used the theory of elasticity of lipid membranes. This allowed us to obtain the dependence of the system’s free energy on the parameters characterizing the membrane, for example, the membrane composition. Thus, we investigated the influence of FFAs on monolayer–bilayer fusion and the likelihood of pexopodia formation, and calculated the energy required for this reorganization. We utilized molecular dynamics simulations to model the fusion process and to verify theoretical results and assumptions.

## 2. Materials and Methods

### 2.1. Formation of the Stalk

We calculated the trajectory of the monolayer–bilayer fusion process, starting from two plane-parallel membranes. We first considered the formation of the stalk resulting from membrane deformation and the appearance of local hydrophobic defects [28]. At small distances (~0.5 nm), these defects attracted each other to reduce the hydrophobic repulsion between membranes [2]. Thus, we considered three main components of the free energy of the system: the membrane elastic deformation energy *W_e_*, the energy of hydration repulsion between the membranes *W_h_* and the energy of interaction between hydrophobic defects *W_f_* [29]. To evaluate the energy barrier analytically, we introduced several simplifying assumptions. We suggested that only the monolayer membrane deformed during stalk formation and divided the process of stalk formation into two stages, while the bilayer remained undeformed as a more rigid object. In the first stage, the monolayer membrane deformed and gave rise to a bulge, called a dimple; as a result, a strong hydration repulsion occurred between this dimple and the bilayer membrane. After that, radially symmetrical hydrophobic defects formed in the monolayer and bilayer, relaxing hydration repulsion between the membranes. We took into account the deformation energy *W*_e_ of the monolayer at both stages. In the first stage, we took into account the energy of hydration repulsion *W_h_* between the membranes, and in the second stage, the interaction energy of hydrophobic defects *W_f_*. Below we describe all three components used in the calculations: *W_e_*, *W_h_* and *W_f_*.

Membrane monolayer deformations were treated in the framework of the Hamm–Kozlov model [30]. To describe monolayer deformations, we introduced a field of unit vectors called directors, **n**, characterizing the average orientation of lipid molecules. The field of directors was defined on a certain surface within the monolayer. The shape of the surface was determined by the unit vectors **N** normal to it (directed towards the intermonolayer surface of the membrane). We took into account two deformation modes—bending and tilt. These deformations were attributed to the so-called neutral surface, where the bending and lateral extension deformations were energetically uncoupled [31]. Bending deformation was quantitatively described by the divergence of the director along the neutral surface, whereas tilt deformation was described by the tilt vector field **t** = **n**/(**nN**) − **N** ≈ **n** − **N**. We assumed the membrane deformation was small; hence, the energy of the deformed monolayer counted from the state of the planar monolayer could be expressed as [32]
(1)$$W=\int \left(\frac{B}{2}{\left(\mathrm{div}n+J_{s}\right)}^{2}-\frac{B}{2}{J_{s}}^{2}+\frac{K}{2}t^{2}+\tilde{\sigma }\right)dS-\sigma A_{0}.$$

Here, *B* and *K* were bending and tilt moduli, respectively, $\tilde{\sigma }$ was the lateral tension of the monolayer, *J_s_* was the monolayer spontaneous curvature, *dS* was the neutral surface area element and *A*_0_ was the area of the neutral surface in the initial unperturbed state.

We introduced the Cartesian system of coordinates. The *Oz* axis was directed perpendicular to the unperturbed bilayer membrane and the *Or* axis was perpendicular to the *Oz* axis and lied in the plane of the neutral surface of the unperturbed monolayer membrane. We assumed an axial symmetry, with *Oz* being the axis of symmetry; this allowed us to consider the dependence of the director and tilt vector only on the radial coordinate. The smallness of deformations meant that the projection of the director on the *Or* axis was much smaller than unity. This allowed us to replace all vector quantities by their projections onto the *Or* axis: **n** → *n_r_* = *n*, **N** → *N_r_* = *N*, **t** → *t_r_* = *t* and div(**n**) → *dn/dr* + *n*/*r*. The tilt vector projection *t* could be related to the director projection *n* by introducing the shape of the monolayer neutral surface *h* using the local volume incompressibility condition [33,34]. As a result, the state of the monolayer was described by two functions—director *n*(*r*) and neutral surface shape *h*(*r*), and the elastic energy density *w_e_* took the following form:(2)$$w_{e}=2\pi r\left(\frac{B}{2}{\left(\frac{dn}{dr}+\frac{n}{r}+J_{s}\right)}^{2}-\frac{B}{2}{J_{s}}^{2}+\frac{K}{2}{\left(n-\frac{dh}{dr}\right)}^{2}+\tilde{\sigma }{\left(\frac{dh}{dr}\right)}^{2}\right).$$

Variation in the energy functional $\int w_{e}dS$ yielded Euler–Lagrange equations, the solution of which allowed us to find explicitly the functions *h*(*r*) and *n*(*r*). They contained free constant coefficients, which were defined from boundary conditions. We divided the membrane into two parts—the dimple of radius *R* and the transition zone from *R* to infinity. Director and neutral surface shape were stitched at point *r* = *R*. The distance between the membranes at infinity was fixed and was taken to be equal to *H*_0_. The distance between the top of the dimple and the neutral surface of the unperturbed monolayer was denoted as *H* (see Figure 2a).

Taking this into account, we obtained the elastic energy *W_e_* of the monolayer as a function of radius *R* and height *H*:(3)$$\begin{array}{l} W_{e}=\pi K\frac{H^{2}R}{l}\sigma^{3/2}\sqrt{1+\sigma }I_{1}\left(R_{\sigma }\right)K_{1}\left(R_{\sigma }\right)\times \\ \times \frac{I_{1}\left(R_{\sigma }\right)K_{0}\left(R_{\sigma }\right)+I_{0}\left(R_{\sigma }\right)K_{1}\left(R_{\sigma }\right)}{{\left(I_{1}\left(R_{\sigma }\right)K_{0}\left(R_{\sigma }\right)+\left(I_{0}\left(R_{\sigma }\right)-1\right)K_{1}\left(R_{\sigma }\right)\right)}^{2}}. \end{array}$$

Here *I* and *K* were modified Bessel functions of the first and second kind, respectively; $l=\sqrt{B/K}$; $R_{\sigma }=\frac{R}{l}\sqrt{\frac{\sigma }{\sigma +1}}$; and $\sigma =\tilde{\sigma }/K$ was the dimensionless lateral tension. For more information on the methodology for calculating elastic energy, please refer to our previous papers [35,36,37].

We then considered the energy of hydration repulsion *W_h_*. It was manifested due to the interaction of water layers oriented near the lipid polar headgroups. The surface energy density of hydration repulsion could be approximated by the formula [12,38]:(4)$$w_{h}=P_{0}\xi_{h}\exp\left(-\frac{z\left(r\right)}{\xi_{h}}\right),$$
where *z*(*r*) was the distance between the membranes as a function of the radial coordinate *r*, *ξ_h_* was the characteristic length of repulsion and *P*_0_ was the pressure at zero separation. Characteristic length *ξ_h_* was ≈ 0.2–0.3 nm and pressure *P*_0_ was ≈ 100–1000 *k_B_T*/nm^3^, both depended on the particular lipid composition of the membrane [39]. Due to the small value of *ξ_h_* and the exponential decay law of energy (4), hydration repulsion was most significant in the dimple region at *r* < *R*. Moreover, the distance between the proximal monolayers increased rapidly with *r*. This allowed us to neglect the hydration repulsion at radii greater than the dimple radius *R*. Integrating the energy density over the area of the dimple and replacing the monolayer shape with the plane, we obtained the following formula for the energy *W_h_*:(5)$$W_{h}=\pi R^{2}\xi_{h}P_{0}\left(\exp\left(-\frac{H_{0}-H}{\xi_{h}}\right)-\exp\left(-\frac{H_{0}}{\xi_{h}}\right)\right).$$

The membrane energy during dimple formation was reduced to the sum of *W_e_* and *W_h_* and depended on the dimple radius *R* and height *H*. We chose *H* as the reaction coordinate changing from zero at the beginning of the fusion to *H*_0_ at the moment of stalk formation. For each value of *H*, the energy was minimized with respect to *R*; as a result, we obtained the minimal energy *W*_1_ of the dimple with height *H* and optimal radius *R^min^*:(6)$$W_{1}\left(H,R^{min}\left(H\right)\right)=W_{e}+W_{h}.$$

When membranes approached each other, the formation of hydrophobic defects became energetically favorable. The energy of interaction between defects in the opposing membranes could be written as [2,40]:(7)$$W_{f}=2\pi \rho^{2}\sigma_{0}\left(1-\exp\left(-\frac{H_{0}-H}{\xi_{f}}\right)\right),$$
where *ρ* was the defect radius. The formation of the defect changed the monolayer elastic energy, as inside the defect the monolayer neutral surface did not exist. Assuming the distance from the defect to the *Or* axis remained equal to *H* (see Figure 2b), the elastic energy *W_e_*_2_ of the remaining membrane became equal to
(8)$$W_{e2}=\pi K\frac{H\rho }{l}\sqrt{\sigma \left(1+\sigma \right)}\left(2J_{s}l^{2}+H\sigma \right)\frac{K_{1}\left(\rho_{\sigma }\right)}{K_{0}\left(\rho_{\sigma }\right)},$$
where *ρ_σ_* was defined similarly to *R_σ_*. We assumed that the defect radius *ρ* equaled the radius of the dimple *R^min^* for each given distance *H*. Thus, the energy associated with the dimple disappeared completely because the dimple was substituted with the hydrophobic defect. As a result, the energy *W*_2_ of the defect as a function of *H* was given by the following equation:(9)$$W_{2}\left(H,R^{min}\left(H\right)\right)=W_{e2}+W_{f}.$$

For such a defect to form, the energy *W*_1_ of the dimple had to be equal to the energy of the defect *W*_2_ at some critical value *H**. This height determined the value of the stalk energy barrier *E* according to the formula *E* = *W*_1_(*H**). Formed defects attracted each other making the mutual approach of the membranes energetically favorable. At the moment of their contact, hydrophobic defects transformed into the stalk, the energy of which was determined by the Formula (8) with *H* = *H*_0_.

### 2.2. Formation of the Bilayer Stalk

We used the proposed model to calculate the energy barrier *E^bil^* towards stalk formation in the case of bilayer fusion to compare it with the barrier *E* in the course of monolayer–bilayer fusion. We estimated *E^bil^* based on simple geometric considerations. Due to the symmetry of the system, deformations of both bilayers were identical. Membranes protruded towards each other; the height of each protrusion was *H* (see Figure 2c). We assumed that the deformational energy of each monolayer of the bilayer was approximately equal to the energy *W_e_*. Thus, the elastic energy of two bilayers was equal to *W_e_^bil^* = 4·*W_e_*. After the opening of the hydrophobic defect, the distal monolayer remained intact, its deformations did not change significantly (see Figure 2d) and the energy was given by *W_e_*. Thus, the elastic energy of the membrane after the defect opening equaled *W_e_*_2_*^bil^* = 2·*W_e_* + 2·*W_e_*_2_. Hydrophobic and hydration energies *W_h_^bil^* and *W_f_^bil^* were calculated as follows:(10)$$\begin{array}{l} {W_{h}}^{bil}=\pi R^{2}\xi_{h}P_{0}\left(\exp\left(-\frac{H_{0}-2H}{\xi_{h}}\right)-\exp\left(-\frac{H_{0}}{\xi_{h}}\right)\right);\\ {W_{f}}^{bil}=2\pi \rho^{2}\sigma_{0}\left(1-\exp\left(-\frac{H_{0}-2H}{\xi_{f}}\right)\right). \end{array}$$

Energies *W*_1_*^bil^* and *W*_2_*^bil^* of the system with dimples and the system with hydrophobic defects were introduced by analogy with the previous case as *W*_1_*^bil^* = *W_e_^bil^* + *W_h_^bil^* and *W*_2_*^bil^* = *W_e_*_2_*^bil^* + *W_f_^bil^*. The height of the barrier was calculated in the same way as for the monolayer–bilayer system. It should be noted that, in the case of monolayer–bilayer fusion, stalk formation corresponded to *H* = *H*_0_, since the bilayer was assumed undeformed; while in the case of bilayer fusion, due to the symmetry of the system, stalk formation corresponded to *H* = *H*_0_/2.

### 2.3. Expansion of the πShaped Structure

We considered the expansion of the stalk leading to an increase in the radius *R* of the contact between the lipid droplet monolayer and the outer monolayer of the peroxisome. The inner monolayer of the peroxisome was located in the lumen of the formed structure, separating the internal contents of the lipid droplet from the peroxisome (see Figure 3).

We estimated the energy required to expand such a π-shaped structure while taking into account the possibility of incorporating FFAs into the membrane. In accordance with the earlier obtained results [27], we assumed that the tilt deformation produced a negligibly small contribution to the energy of the monolayer, and we took into account only the bending deformation mode. As before, we assumed that the inner monolayer of the peroxisome was not deformed, and all the elastic energy change was associated with the bending of the outer monolayer. The membrane was most strongly curved in the transition zone between the lipid droplet and the peroxisome (colored with green in Figure 3b). There were two contributions to the mean curvature of the membrane: the equatorial and the meridional principal curvatures. The equatorial curvature was determined by the radius of the π-shaped structure, $J_{e}\approx \frac{1}{R}$, and the meridional curvature was determined by the distance *H*_0_ between the membranes and had the opposite sign: $J_{m}\approx -\frac{1}{H_{0}}$. The mean curvature was given by the sum of these two contributions: $J∼\frac{1}{R}-\frac{1}{H_{0}}$. The energy density of the membrane was determined by the total curvature, which included the spontaneous curvature of the membrane: $w_{bend}=\frac{B}{2}{\left(J-J_{s}\right)}^{2}-\frac{B}{2}{J_{s}}^{2}$. We assumed that the membrane between the peroxisome and the lipid droplet was approximately cylindrical; as a result, the bending energy of the membrane took the following form:(11)$$W_{\pi }=2\pi RH_{0}\frac{B}{2}\cdot \left({\left(\frac{1}{R}-\frac{1}{H_{0}}-J_{s}\right)}^{2}-{J_{s}}^{2}\right).$$

We assumed that FFAs could penetrate the most stressed membrane area, relaxing the elastic energy. This process was modeled as a change in the spontaneous curvature *J_s_*, which led to a change in the asymptotical behavior of the energy (11) at large radii *R* → ∞.

### 2.4. Molecular Dynamics

We performed a coarse-grained molecular dynamic (MD) simulation to study bilayer–bilayer and bilayer–monolayer stalk formation. The simulation was performed with the Martini 2.2 coarse-grained force field [41]. The cut-off for Lennard–Jones and Coulomb interactions was set to 1.1 nm. The Verlet scheme was used to update the neighbour list. To keep the temperature constant at 300 K and pressure at 1 bar, a velocity rescaling thermostat [42] and Berendsen barostat [43] were employed. We used a separate temperature coupling group for each water layer, bilayer or monolayer with contact with triglycerides (TGs). The time step for the MD simulation was set to 20 fs. MD simulations were carried out with a modified GROMACS 2018.8 [44], which used a specially designed reaction coordinate *ξ*_ch_ [45,46]. This so-called “chain coordinate” *ξ_ch_* described the interaction of two monolayers in a specific volume. This volume was defined as a cylinder connecting monolayers and was divided into 1 Å thick slices. The *ξ_ch_* coordinate was determined by the fraction of the slices filled with tails of the lipid monolayers:(12)$$\xi_{ch}=\frac{1}{N_{s}}\sum_{s=0}^{N_{s}-1}\delta_{\varsigma }(n_{s}^{\left(t\right)})$$

Here, $N_{s}$ was the total number of cylinder slices, $n_{s}^{\left(t\right)}$ was the number of tail beads in slice *s* and $\delta_{\varsigma }$ was a (differentiable) indicator function that took $\delta_{\varsigma }=0$ for an empty slice ($n_{s}^{\left(t\right)}=0$) and $\delta_{\varsigma }=1$ for a filled slice ($n_{s}^{\left(t\right)}>0$).

To obtain the energy of bilayer–bilayer and bilayer–monolayer stalk formation, we computed the potential of mean force (PMF) along the reaction coordinate *ξ_ch_*. The initial configuration of the system was built by combining dioleoylphosphatidylcholine (DOPC) lipid bilayers obtained with the CHARMM-GUI web service [47,48]. For the bilayer–monolayer simulation one of the membranes was split into two monolayers and the void between the monolayers was filled with TG molecules. The resulting systems contained a lipid droplet with a ~4 nm layer of TGs. The middle water layer contained 1390 water beads which corresponded to a ~2 nm distance between the phosphate group of opposite monolayers. Umbrella sampling with 36 windows was used to compute the PMF. Windows were equally spaced between *ξ_ch_* = 0.1 and *ξ_ch_* = 1, and the 3000 kJ/mol force constant was used to keep the system at a specific value of reaction coordinate. Each window was equilibrated for 50 ns before the production run of 150 ns. The weighted histogram analysis method (WHAM) [49] was implemented as the gmx WHAM module [50] was employed to compute PMFs.

## 3. Results

### 3.1. Energy Barrier for Stalk Formation

To find the energy barrier for stalk formation *E*, we calculated the energy depending on the distance *H* between the membranes according to the model presented in Section 2.1. The height of the energy barrier depended on the combination of the material parameters characterizing the membrane. In particular, barrier height *E* depended on the lipid composition of the membrane, therefore, this dependence could be used to verify the used model. The membranes under study mainly consisted of DOPC (dioleoylphosphatidylcholine) and DOPE (dioleoylphosphatidylethanolamine); DOPC being the main component. Because of this, we first used the typical parameters of the DOPC membrane and then studied the effect of DOPE addition to the mixture. We used the following values of the energy parameters: bending modulus *B* = 10 *k_B_T* [51]; tilt modulus *K* = 10 *k_B_T*/nm^2^ [52]; spontaneous curvature *J_s_* = −0.1 nm^−1^ [53]; dimensionless lateral tension *σ* = 0.001, which corresponded to the lateral tension $\tilde{\sigma }$ being equal to 0.04 mN/m = 0.01 *k_B_T*/nm^2^ [29]; hydration repulsion parameters *P*_0_ = 800 *k_B_T*/nm^3^ and *ξ_h_* = 0.235 nm [39]; and hydrophobic interaction parameters *σ*_0_ = 12.5 *k_B_T*/nm^2^ and *ξ_f_* = 1 nm [2]. The value of the fixed distance *H*_0_ between the membranes was assumed to be equal to 3 nm. As a result, we obtained the dependence of the energies *W*_1_ and *W*_2_ on the reaction coordinate *H*, which are shown in Figure 4a. To compare the obtained results with the case of bilayer fusion, we also calculated the corresponding energy trajectories, which are shown in Figure 4b.

Energy dependencies shown in Figure 4 allowed us to calculate the height *E* of the energy barrier as *E* = *W*_1_(*H**), where *H** was the height of transition from the dimple to the hydrophobic defect. According to our calculations, the barrier height was equal to *E* = 35 *k_B_T* in the case of monolayer–bilayer fusion and *E^bil^* = 41 *k_B_T* in the case of bilayer fusion. In the next step, we validated our model by varying the DOPE content in the membrane, which affected the values of parameters *J_s_*, *P*_0_ and *ξ_h_*. We also varied the distance *H*_0_ between the membranes, which modeled different initial conditions for membrane fusion due to the action of tethering proteins or due to the result of a balance of repulsive hydration and attractive van der Waals forces. The second case allowed us to consider the possibility of protein-free membrane fusion. We assumed that the distance *H*_0_ could vary from 2 to 4 nm [13]. The dependencies of barrier heights *E* and *E^bil^* of monolayer–bilayer and bilayer fusion on *H*_0_ are shown in Figure 5 for cases of pure DOPC and the mixture of DOPC:DOPE = 1:1. In the latter case, parameters *J_s_*, *P*_0_ and *ξ_h_* were equal to *J_s_* = –0.25 nm^−1^, *P*_0_ = 60 *k_B_T*/nm^3^ and *ξ_h_* = 0.37 nm [39,53].

The addition of DOPE to the membrane reduced the barrier height, both in the cases of monolayer–bilayer and bilayer fusion. The values of *E* and *E^bil^* were lowered by several units of *k_B_T*, depending on *H*_0_. A decrease in the barrier height was associated with both a decrease in the spontaneous curvature value and a change in the parameters of hydration repulsion *P*_0_ and *ξ_h_*. Thus, the barrier height value *E* decreased from 35 *k_B_T* to 32 *k_B_T* in the case of the DOPC:DOPE 1:1 mixture with *H*_0_ = 3 nm, and *E^bil^* decreased from 41 to 37 *k_B_T* with the same value of *H*_0_.

We also measured the free energy barrier for bilayer–bilayer and bilayer–monolayer stalk formation utilizing molecular dynamics simulations. To this end, we used umbrella sampling to calculate the dependence of the system energy Δ*G* on the reaction coordinate *ξ_ch_* in both cases. The results of these calculations are depicted in Figure 6.

The energy barrier for stalk formation was measured as a difference between the maximum and minimum values of the free energy profiles in Figure 6. The energy of stalk formation for bilayer–bilayer fusion was equal to 44 ± 1 *k_B_T*; the replacement of one bilayer with the monolayer contacting with TGs reduced the energy barrier to 33 ± 1 *k_B_T*. The shapes of the energy profiles differed qualitatively for these systems. In the case of bilayer–monolayer fusion, the resulting configuration lied in a small energy minimum of ~1.5 *k_B_T*, while no energy minimum was observed for bilayer–bilayer stalk formation. Thus, in the first case, the stalk was a metastable configuration in contrast to bilayer–bilayer fusion.

To obtain insight into the shape of the stalk, we computed the number density of the phosphate group. The resulting density maps are shown in Figure 7.

The density maps were calculated from the 100 ns trajectory with constant chain coordinate *ξ_ch_* = 1. The density map of the stalk formed by bilayers was symmetric and had a clear outline in the stalk region. Contrary to this, the stalk formed by the bilayer and monolayer had an asymmetric profile deformed towards the bilayer. Moreover, the density in the stalk region was more vague and did not have a clear contour, which indicated the more dynamic stalk structure for the bilayer–monolayer system.

### 3.2. Energy of the π-Shaped Structure

To investigate the behavior of the system after stalk formation, we calculated the energy *W_π_* of the π-shaped structure depending on the radius *R* according to the Formula (11). This expression could be considered as an approximation of the energy for sufficiently large radii *R* > *H*_0_ when the equatorial curvature could be considered constant and zero. The spontaneous curvature *J_s_* of the membrane could be changed by modifying the transition zone between the membranes with FFAs, such as oleic acid (OA). The membrane spontaneous curvature should have decreased upon insertion of FFAs since they increased the volume of the hydrocarbon region of the membrane, keeping the headgroup region almost intact. According to that, we calculated the energy *W_π_* dependence on radius *R* for the values of *J_s_* = −0.1 nm^−1^, *J_s_* = −0.15 nm^−1^ and *J_s_* = −0.2 nm^−1^. These dependencies are shown in Figure 8. Note that the graphs presented describe the behavior of the π-shaped structure in the case of large values of *R*, while stalk formation occurred at a small radius ~1–2 nm. Earlier, the case of intermediate values of the radius was considered in work [27], and it was shown that the transition to large radii was associated with overcoming a small energy barrier of ~5–10 *k_B_T* height. We believed that this barrier was negligible and did not affect the state of the system.

As could be seen from Figure 8 and expression (11), the system’s behavior was determined by a combination of the spontaneous curvature and the intermembrane distance *H*_0_. At large radii, the energy of the system had a linear asymptotic behavior:(13)$$W_{\pi }∼\frac{\pi B}{H_{0}}\left(2H_{0}J_{s}+1\right)R.$$

According to the Equation (13), expansion of the π-shaped structure became energetically favorable, when the spontaneous curvature *J_s_* < −1/2*H*_0_. If distance *H*_0_ was of the order of 2–3 nm, then the spontaneous curvature had to be lower than −0.17–−0.25 nm^−1^ to make the expansion favorable. If the membrane consisted of pure DOPC, its spontaneous curvature was equal to ~−0.1 nm^−1^, and in this case, the expansion was energetically unfavorable. At the same time, the incorporation of FFAs into the membrane led to a decrease in the spontaneous curvature, facilitating the expansion of the π-shaped structure. The incorporation of FFAs also led to an increase in the area of the membrane of lipid droplets with their volume being practically unchanged. As a result, excess membrane area sagged in the center of the π-shaped structure and formed pexopodia. Note that the negative spontaneous curvature of oleic acid led to the bending of the pexopodia membrane towards the lipid droplet interior, and not outwards, in accordance with the observed experimental tendency [26]. Further uncompensated FFAs incorporation may have resulted in enlargement of the area in the form of tubules; this is a convenient adjustment mechanism since the tubules have virtually zero volume with a sufficiently large surface area.

## 4. Discussion

In the present work, we calculated the trajectory of the monolayer–bilayer fusion process that occurs upon the contact of a lipid droplet and a peroxisome. This process could be divided into two stages: the formation of the stalk and its expansion into the π-shaped structure leading to pexopodia formation. The first stage seemed to be the most difficult for theoretical modelling due to the interaction between membranes and their strong deformation. To this end, we developed a simplified model that restricted the intermembrane interaction to a very small region and assumed the bilayer membrane to be undeformed due to its higher rigidity compared to the lipid droplet monolayer. This allowed us to significantly simplify the calculation procedure and compute the height of the energy barrier *E* of stalk formation. To validate our model, we applied a similar calculation procedure for the case of bilayer fusion, estimated the barrier values for bilayer stalk formation *E^bil^* and showed that they agreed with the existing literature data.

This energy barrier decreased with the addition of components with negative spontaneous curvature, which relieved the elastic stress of the stalk, fitting the negative mean curvature of this structure. Thus, the addition of 50% DOPE could drop the barrier by ~20 *k_B_T* [44,54], qualitatively agreeing with our results. The decrease in the initial intermembrane distance *H*_0_ reduced the deformations required for the approaching of the membranes, and led to the dehydration of the proximal lipids, facilitating the opening of the hydrophobic defect. Thus, a decrease in the *H*_0_ from 2.5 to 1 nm could drop the energy barrier by ~20 *k_B_T* [44], which agrees with our findings (Figure 5). Energy barriers obtained in molecular dynamics simulations also agreed with our theoretical estimations. Thus, the generalization of our model to the case of bilayer fusion, despite its simplifications, gave results quantitatively consistent with the literature data and could be considered as validation of our model.

In the case of monolayer–bilayer fusion, the outlined above energy barrier dependencies also appeared to be valid. The obtained results showed that the barrier to stalk formation in monolayer–bilayer fusion was smaller by several units of *k_B_T* compared to that of bilayer fusion (see Figure 5). Theoretical calculations yielded the value of 25 *k_B_T* for the barrier height when *H*_0_ = 2.5 nm, which was 4 *k_B_T* lower than the *E^bil^* value since the deformation of the bilayer involved more deforming objects than the monolayer one. Molecular dynamics calculations confirmed the theoretical results (see Figure 6)—the monolayer–bilayer fusion barrier equaled 33 *k_B_T*, which was 10 *k_B_T* lower than the bilayer fusion barrier. The density distribution of lipid polar heads also revealed the asymmetry of monolayer and bilayer deformations (see Figure 7), which confirmed the initial assumption of our model. At the same time, molecular dynamics calculations indicated an important feature of monolayer–bilayer fusion. A comparison of energy trajectories in Figure 6 signified greater stability of the stalk intermediate for monolayer–bilayer fusion, since at the end of the trajectory (chain coordinate *ξ_ch_*~1) energy was decreasing, indicating possible stalk metastability. In the case of bilayer fusion, the energy monotonously increased until the point of the membrane contact. Similar behavior for bilayer fusion of pure DOPC membranes was also observed in the work of Poojari and coworkers [44]. This result implies that a monolayer–bilayer stalk can be formed without the direct assistance of fusion proteins since it is energetically more stable than a bilayer stalk. Even in the case of the pure DOPC membrane, the barrier height was in a range from 25 to 30 *k_B_T*; this barrier can be overcome in a biologically reasonable time with thermal fluctuations alone.

The formation of a stalk signifies the initial contact of merging membranes, resulting in the appearance of a π-shaped structure. For the formation of pexopodia, this structure must expand until its radius is of the order of 100 nm. Our results indicated that expansion of the π-shaped structure was energetically favorable or unfavorable depending on the spontaneous curvature of the transition zone between the fusing membranes (colored in green in Figure 3b). This region was approximated as a toroid with a fixed radius *R*. When the radius *R* became large enough, the toroid net curvature became negative and was determined by the distance *H*_0_ between the membranes. If the spontaneous curvature of the membrane in the transition zone was sufficiently negative, it compensated for the bending of the monolayer, and the expansion of the structure became energetically favorable. Our estimates showed that the spontaneous curvature of a pure DOPC membrane was not negative enough to compensate for the bending energy of the monolayer of the assumed geometrical shape. If the distance *H*_0_ between the membranes was approximately 2.5 nm, the spontaneous curvature had to be –0.2 nm^−1^, while the spontaneous curvature of DOPC was –0.1 nm^−1^. This meant that for a successful π-shaped structure opening, the monolayer must have contained a large admixture of components with significant negative curvature. We assumed that this role could be played by oleic acid, resulting from the breakdown of the neutral lipids in the core of the lipid droplet. This fatty acid had a sufficient spontaneous curvature; according to the work of Gillams and coworkers [55], a mixture of DOPC:OA = 1:1 has a spontaneous curvature of –0.4 nm^−1^. This value is by far sufficient for the relaxation of the elastic energy of the π-shaped structure. For this to happen, oleic acid must have preferentially distributed into the transition zone between the membranes. This result was in line with the previous molecular dynamics calculations [44,54], indicating the predominant distribution of the component with negative curvature into the region of the stalk.

The expansion of the structure continued until its radius became on the order of several tens of nanometers. After this threshold, the elastic energy became very small due to the low curvature and the behavior of the system was determined by the surface energy. We assumed that at this stage the state of the surface was determined by the incorporation of FFAs, such as oleic acid, into the area of contact of the monolayers, which led to an increase in the membrane area and further energy relaxation. This made it possible to explain the experimentally observed behavior of the system, in particular, the appearance of pexopodia and their protrusion towards the core of the lipid droplet. The latter was a consequence of the negative spontaneous curvature induced by oleic acid in the previous step. This assumption also allowed us to explain the appearance of the so-called “gnarls” [26] as a result of an excessive increase in membrane area in the absence of enzymes that break down FFAs in the peroxisome.

Our results indicate a fundamental difference between the energy trajectory of monolayer–bilayer fusion and the classical case of bilayer fusion. The latter required overcoming at least two [56] or three [13] energy barriers until complete membrane fusion. At the same time, successful monolayer–bilayer fusion required overcoming only one energy barrier, which was lower than the corresponding barrier in bilayer fusion. This demonstrates the ease of monolayer–bilayer fusion and the possibility of its occurrence without the direct participation of fusion proteins. The question arises why lipid droplets do not fuse with other organelles and do not form structures similar to peroxisomes. We attributed this specificity to the incorporation of FFAs into the stalk region, which lowered the elastic energy of the system and contributed to the expansion of the π-shaped structure. Without this incorporation, fusion ended at the moment of stalk formation and π-shaped structure expansion was energetically unfavorable. Thus, the breakdown of the lipid droplets’ core neutral lipids into FFAs was necessary for the formation of pexopodia, which is in line with experimental data [26]. To the best of our knowledge, the mechanism that triggers this breakdown has not yet been precisely established. Following Binns and coworkers [26], we assumed that fat breakdown was triggered by the initial contact of the lipid droplet with the peroxisome and, thus, ensured the specificity of fusion. Further research in this area and the precise determination of tethering proteins for lipid droplet–peroxisome contact could help to verify this hypothesis.

## 5. Conclusions

In this work, we studied the process of fusion of the lipid monolayer and bilayer and compared it with bilayer–bilayer fusion. We developed the trajectories of these processes using the continuum elasticity theory of lipid membranes and investigated the details of fusion structures utilizing molecular dynamics. We considered two stages of the monolayer–bilayer fusion process: the initial contact of the membranes leading to the formation of a stalk and its consequent radial expansion into the π-shaped structure. We found that in the given system, the stalk was energetically stable and had a lower energy barrier of formation as compared to the case of bilayer fusion. The considered energy barrier decreased with an increase in DOPE content in the membrane and with a decrease in the distance between the membranes, which coincided with the trends in the case of bilayer fusion. The further evolution of the monolayer–bilayer stalk depended qualitatively in a threshold manner on the membrane composition. Its expansion became energetically viable only if the membrane spontaneous curvature was lower than ~−0.2 nm^−1^. In the specific case of interaction between the peroxisome and the lipid droplet, this requirement could be met by incorporation of free fatty acids possessing negative spontaneous curvature into the stalk region. This incorporation led to the radial expansion of the stalk to the experimentally observed structures, called pexopodia. This picture suggested the mechanism of involvement of the fat lysis reaction in the process of fusion of a peroxisome with a lipid droplet. Transformation of the fat into the free fatty acids in the lipid droplets induced negative curvature in the membrane, enhancing its fusion with the peroxisome, which was required for the subsequent β-oxidation of the free fatty acids.

## Figures and Tables

**Figure 1 membranes-12-00992-f001:**
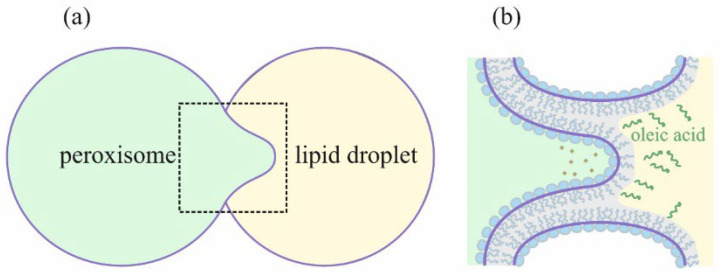
(**a**) Schematic representation of the peroxisome–lipid droplet membrane fusion. The region of membrane contact is marked with a dotted square. (**b**) Schematic representation of the pexopodia structure. Free fatty acids are denoted with green color. Beta-oxidizing enzymes are denoted as diamonds.

**Figure 2 membranes-12-00992-f002:**
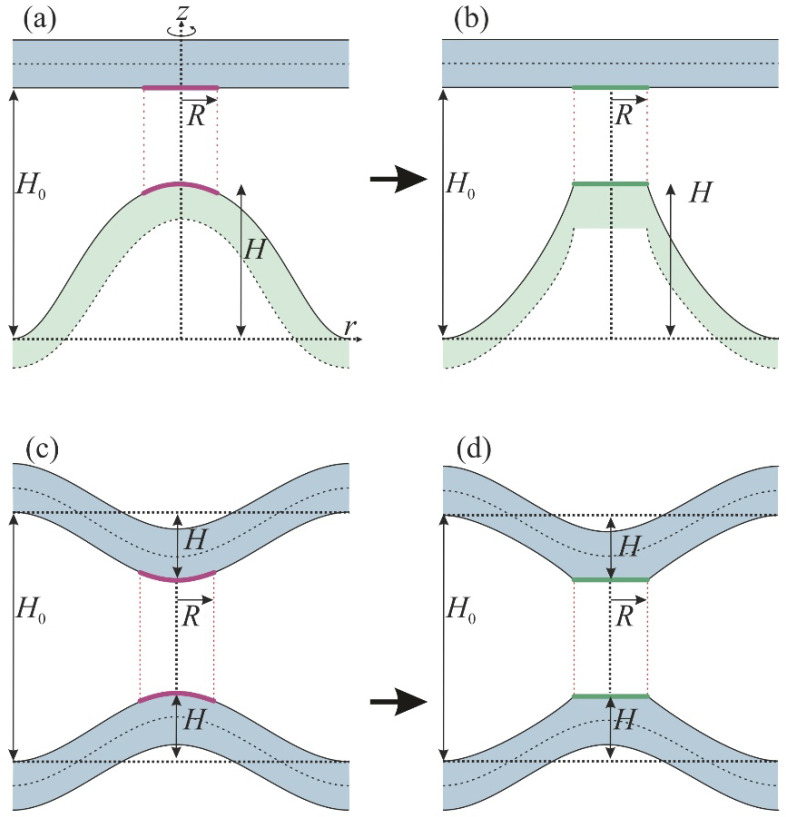
Schematic representation of the membrane remodeling prior to stalk formation in the cases of monolayer–bilayer (panels (**a**,**b**)) and bilayer–bilayer (panels (**c**,**d**)) fusion. The initial distance between the membranes is equal to *H*_0_, the height of the monolayer dimple is denoted by *H* and the total height of bilayer dimples in (**c**,**d**) is 2*H*. Radius *R* of the dimple limited the effective area of hydration repulsion, denoted by crimson on panels (**a**,**c**), and hydrophobic interaction, denoted by green on panels (**b**,**d**). The bilayer membrane is colored blue-green and the monolayer membrane is colored ghostly green. Note the axial symmetry of the system.

**Figure 3 membranes-12-00992-f003:**
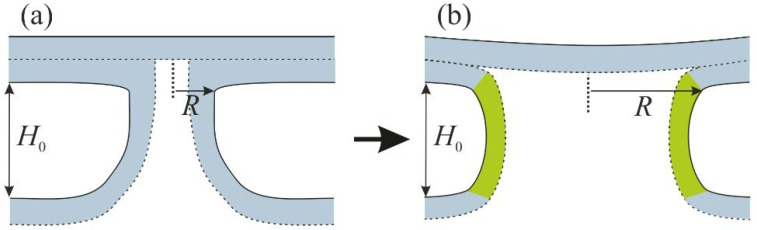
Schematic representation of the π-shaped structure model. (**a**) Origination of the π-shaped structure after stalk formation. (**b**) Expansion of the π-shaped structure; transition zones between the fusing membranes are colored in green. Other parts of the lipid membrane are highlighted in blue. *R* is the structure radius; *H*_0_ is the fixed distance between the membranes.

**Figure 4 membranes-12-00992-f004:**
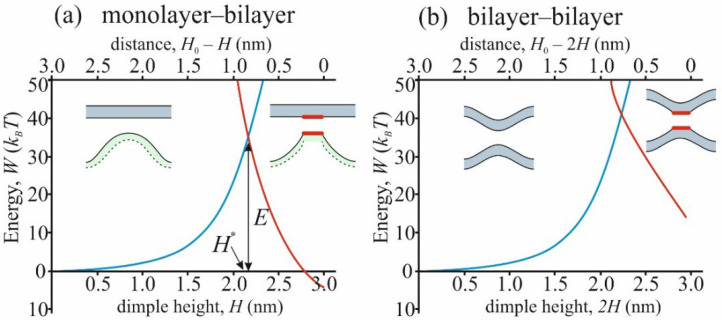
(**a**) The energy profile of monolayer–bilayer fusion. The dependence of the energy of the monolayer with a dimple (*W*_1_, blue curve) and system with opened hydrophobic defects (*W*_2_, orange curve) on the distance between the membranes (*H*_0_ − *H*). (**b**) The energy profile of bilayer–bilayer fusion. The dependence of the energy of the bilayers with a dimple (*W*_1_*^bil^*, blue curve) and system with opened hydrophobic defects (*W*_2_*^bil^*, orange curve) on the distance between the membranes (*H*_0_ − 2*H*). Green color on the insets denotes the monolayer membrane, blue color denotes the bilayer membrane and red lines denote the hydrophobic patches.

**Figure 5 membranes-12-00992-f005:**
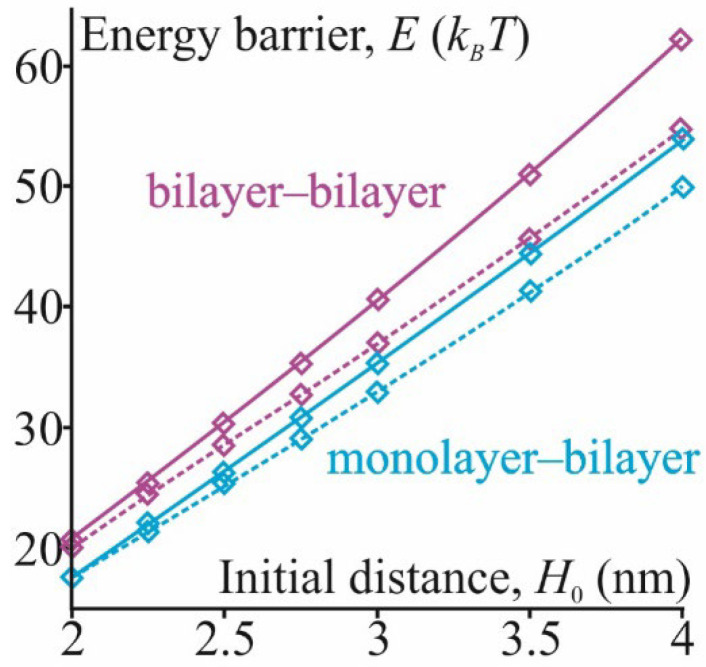
Dependence of energy barriers *E* and *E^bil^* (in *k_B_T*) on the initial distance *H*_0_ for the cases of *Js* = −0.1 nm^−1^, *P*_0_ = 800 *k_B_T*/nm^3^ and *ξ_h_* = 0.235 nm and *Js* = −0.25 nm^−1^, *P*_0_ = 60 *k_B_T*/nm^3^ and *ξ_h_* = 0.37 nm. The first set of parameters model the pure DOPC membrane, and the second, the mixture of DOPC:DOPE = 1:1. Barriers are depicted as follows: cyan color solid line—bilayer–monolayer fusion, pure DOPC; cyan color dashed line—bilayer–monolayer fusion, DOPC:DOPE = 1:1; purple color solid line—bilayer fusion, pure DOPC; and purple color dashed line—bilayer fusion, DOPC:DOPE = 1:1.

**Figure 6 membranes-12-00992-f006:**
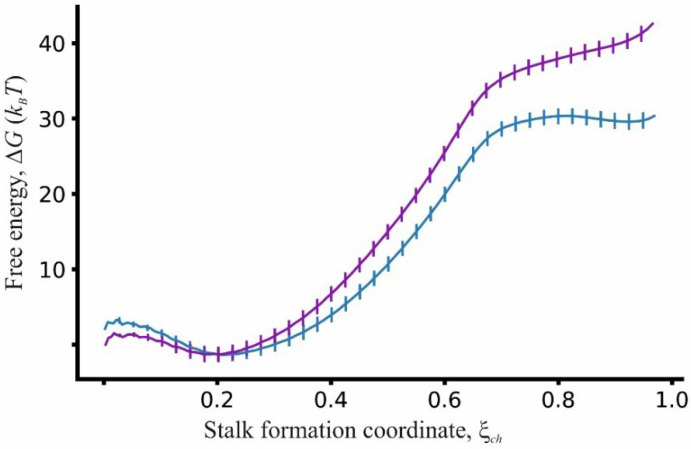
Dependencies of the free energy Δ*G*, measured in the units of *k_B_T*, on the stalk formation coordinate *ξ_ch_* for pure DOPC membranes. The energy of the bilayer–bilayer system is plotted as a purple curve and the energy of the bilayer–monolayer system is plotted as a cyan curve.

**Figure 7 membranes-12-00992-f007:**
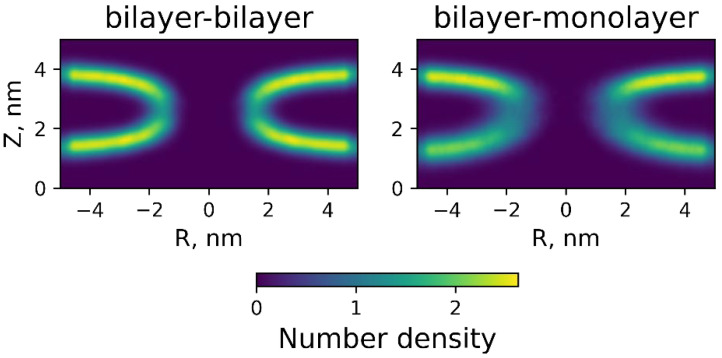
The number density of the phosphate group at lipid stalk for bilayer–bilayer (**left**) and bilayer–monolayer (**right**) systems.

**Figure 8 membranes-12-00992-f008:**
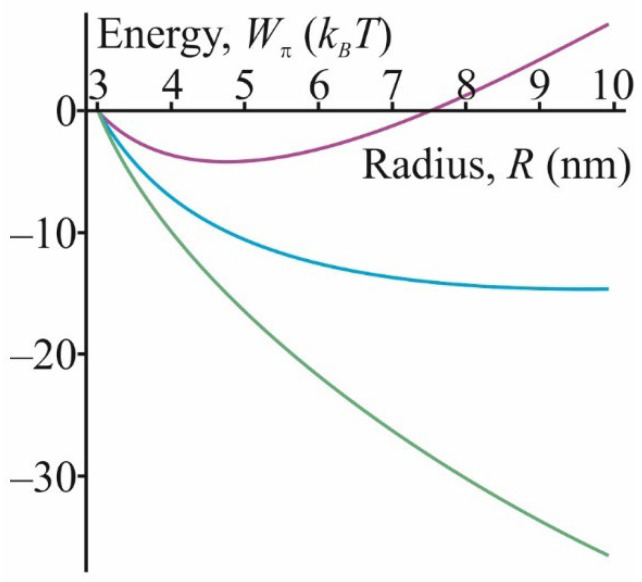
The dependence of the energy of π-shaped structure *W_π_* on its radius *R* for different values of the spontaneous curvature *J_s_* = −0.1 nm^−1^ (purple color), *J_s_* = −0.15 nm^−1^ (cyan color) and *J_s_* = −0.2 nm^−1^ (green color).

## Data Availability

Data available on request from mkalutskiy@inbox.ru.

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
