# Peer review of "A Model of Lipid Monolayer–Bilayer Fusion of Lipid Droplets and Peroxisomes"

_membranes, 2022, doi:10.3390/membranes12100992_

Round 1

Reviewer 1 Report

The paper was well organized and smooth to read. I just have one question, regarding the CGMD, it is important to make sure the simulated length is validated to be enough. Does the author have any metric to measure the system that is converged with the largely-fluctuating NPT ensemble? Usually, it is not easy to do so. Is 50 ns long enough?

Reviewer 2 Report

In my opinion, the submitted manuscript „ Intermediates of Lipid Monolayer–Bilayer Fusion are Energetically Stable and Promoted by Free Fatty Acid Interdigitation” meets aims and scope of „Membranes” Journal and may be accepted after minor revision.

1.       In my opinion, the title of the article takes the form of a sentence, which may not be reader-friendly. Maybe changing the form of the title to: „A model of lipid monolayer–bilayer fusion” or „A model of lipid monolayer–bilayer fusion of lipid droplets and peroxisomes”, or „A model of lipid monolayer–bilayer fusion promoted by free fatty acid interdigitation” will be easier to understand. Consider changing the title.

2.       In my opinion, the publication lacks a short and clear explanation why the authors decided to develop a mathematical model of lipid monolayer-bilayer fusion and how it can be used in practice.

3.       Figure 1 shows an enlarged slice of a peroxisome fused to a lipid droplet. It is difficult to imagine what is what in this picture, especially since the caption below the figure says that both peroxisome and lipid droplets are on the left. Maybe it is better to sign these elements on the drawing, or modify the picture so that it would be easier to imagine what is discussed in the article.

4.       In the sentence: „In this work, we consider…” (line112) it is good to recall Figure 3, which shows what the π–shaped structurer looks like.

5.       In my opinion, when referring to specific works of other scientists, surnames can be cited, not just bibliographic numbers (e.g. line 76, 471, 494, 528).

6.       In the first equation, there is no space after the parentheses.

Round 2

Reviewer 1 Report

Accept in its current form